# The Possibilities of Personalized 3D Printed Implants—A Case Series Study

**DOI:** 10.3390/medicina59020249

**Published:** 2023-01-28

**Authors:** Selim Safali, Till Berk, Biser Makelov, Mehmet Ali Acar, Boyko Gueorguiev, Hans-Christoph Pape

**Affiliations:** 1Orthopaedics and Traumatology Department, Medical Faculty, Selçuk University, Konya 42250, Turkey; 2AO Research Institute Davos, 7270 Davos, Switzerland; 3Department of Trauma, University Hospital Zurich, 8091 Zurich, Switzerland; 4University Multiprofile Hospital for Active Treatment ‘Prof. Stoyan Kirkovitch’, Trakia University, 6003 Stara Zagora, Bulgaria

**Keywords:** three-dimensional printed prostheses, custom made implants, personalized prostheses, surgical treatment

## Abstract

*Background and Objectives*: Following the most recent software and 3D printing developments, the use of personalized 3D printed orthopedic implants for treatment of complicated surgical cases has gained more popularity. Today, orthopedic problems that cannot be solved with standard implants may be effectively addressed using personalized prostheses. The aim of this study is to present the designing, modeling and production stages of four different personalized 3D printed prostheses and their application in clinical cases of patients who underwent treatment in various anatomical locations with a precisely specified indication for implantation. *Materials and Methods*: Based on computed tomography scanning, personalized 3D printed prostheses were designed, produced and used in four patients within a period of three to five days after injury or admission. *Results*: Early term follow-ups demonstrated good to excellent results. *Conclusions*: Personalized 3D printed prostheses offer an opportunity for a treatment of choice and provide good anatomical and functional results, shortened surgical time, less complications, and high satisfaction in patients with appropriate indications. The method should be considered primarily for patients with large bone defects, or such indicated for resection. Personalized 3D printed prostheses have the potential to become more common and beneficial in the future.

## 1. Introduction

Following the most recent software and 3D printing developments, the use of personalized orthopedic implants for treatment of complex surgical cases has gained more popularity. The modern 3D printing technology has contributed to (1) better understanding of complex bone fracture patterns, (2) standardization of the surgical procedures, and (3) proper implant positioning in preclinical studies [1,2,3].

In orthopedic practice, diverse implants are available for standardized surgeries related to bone substitution in various anatomical locations of the human body; however, they are not able to provide appropriate solutions and solve problems in some non-traditional situations when patients’ bony geometries are outside of the range of standard implant applications with respect to implant size and/or disease-specific requirements [4,5]. Salvage surgeries and arthrodesis are indicated in such cases, with a low level of success and patient satisfaction, especially in hip surgery [6]. Solutions can be offered using custom-made and tailored implants. Çıtak et al. reported good results following the application of patient-specific 3D printed prostheses designed for acetabular defects [7]. 

On the other hand, the increasing availability of different imaging modalities along with the advances in analytics and navigation tools have led to development, maturing and wider clinical applications of computer-assisted orthopedic surgery [8,9,10,11,12,13,14,15]. The latter implements approaches with the use of computer-enabled tracking systems or robotic devices, improves the visibility to the surgical field, increases the accuracy in a variety of surgical procedures, and enhances the treatment of musculoskeletal diseases in orthopedics and traumatology. 

Moreover, patient-specific implants can be designed based on medical imaging with the use of specialized software packages allowing images of the patient’s anatomy to be merged with computer-aided design models of the corresponding implant systems. Beyond geometrical considerations to fit the anatomy under consideration of the anatomical landmarks, the biomechanical behavior of fixation constructs can be investigated virtually, predicted and optimized via finite element modeling that can incorporate the mechanical properties of bones, articular surfaces, muscles, tendons, ligaments and implants, and simulate various loading scenarios to evaluate fixation stability and failure risks [16,17,18,19,20,21,22].

Custom-made 3D printed prostheses can be useful for surgical treatment of bone defects in various anatomical locations or in cases with anatomical variations where standard implants cannot be used [23,24,25,26,27,28]. Depending on the patient’s condition, infection rates can be reduced by covering the surface of the designed implants with silver ions [29]. Furthermore, the osseointegration process can be accelerated by applying a surface coating with hydroxyapatite on the bone–implant contact zones [30]. A metal–polyethylene interface can be used in implants designed for joint replacement [31].

Since custom-made implants are tailored to the patient’s specific anatomy, they can fit perfectly into the bone defect, are easy to implant intraoperatively and therefore can significantly improve the affected function. Thus, the operative time is shortened, and subsequent complications can be prevented [7]. As personalized implants provide an adequate anatomical fit without disturbing the load distribution and bone biomechanics, high patient satisfaction is achieved [32,33,34,35,36,37]. Moreover, the modern design of a 3D printed personalized implant not only has the structural geometry that can match the surgical requirements of an individual patient but also allows biomechanical evaluation under patient-specific loading conditions for design modification prior to the actual implant fabrication and application to reduce pain, recovery time and enhance osseointegration—if necessary—and function [38,39].

The design of personalized implants depends on both the surgeon’s knowledge and patient’s needs. Besides the necessity to discuss the patient’s problem and understand the implant requirements with regard to stability and function, the workflow for 3D printing requires implementation of multiple separate engineering software packages to master several steps of the working process under surgeon’s supervision [5]. The large number of available materials, such as metallic alloys, polyetheretherketone [26,40], polycaprolactone, bioceramics and ceramic scaffolds—each one with specific biomechanical and biocompatibility features—allows personalized implant designing to meet the patient’s needs [41,42]. The material selection may vary depending on the anatomical location [23], desired biomechanical strength or need for biodegradability. Titanium alloys are preferred in load-bearing regions such as the acetabulum, femur or tibia, whereas bioresorbable materials are selected for surgeries in the facial region of the skull [28,43]. Knowledge of the anatomy, function, biomechanics, surgical approach and material engineering is required for these important aspects in material design.

The aim of this study is to present the designing, modeling and production stages of four different personalized 3D printed prostheses and their application in clinical cases of patients who underwent treatment in various anatomical locations with a precisely specified indication for implantation.

## 2. Materials and Methods

Ethical approval for this study was obtained from the Ethics Committee of Selçuk University (2022–242). Informed consent was obtained from all participants in the study with characteristics summarized in Table 1.

The workflow for the medical imaging, designing, prototyping, production and surgical application of the personalized 3D printed implants included different stages (Figure 1). To obtain the 3D data, thin-section topographies of the affected and the contralateral side were generated via computed tomography (CT) scanning (SOMATOM Emotion 6 CT scanner, Siemens Healthcare GmbH, Erlangen, Germany). The data were obtained in the Digital Imaging and Communications in Medicine (DICOM) format, then converted into the stereolithography (STL) format and reconstructed into a 3D image, while the contralateral side was converted to a mirror image using Amira software package (V.6.0.0, FEI Company, Hillsboro, OR, USA). The anatomical dimensions of the affected region were reconstructed by overlaying the two data sets. 

Subsequently, the geometry of the planned implant was designed under consideration of the recipient bone site dimensions and the corresponding characteristics of the prototype bone–implant construct. When the implant was planned for placement in the medullary canal, a pedicle-style stem handle was added after medullary canal measurements. In case of a plate-like implant structure, its design, geometry and screw holes positions were adjusted. Further, the prototype construct comprising both implant and surgical site (affected bone) models was manufactured by means of 3D printing (Ultimaker B.V., Utrecht, Netherlands) from a polylactide (PLA) polymer filament to test the proper fitting of the implant prototype to the bone recipient site in a mock surgery. If the designed prototype was compatible with the location of its application, additional attachments, screws and extensions were considered. Suture holes were planned if soft tissues needed to be sutured to the implant. When the implant prototype or its attachments did not completely match the recipient site, redesigning was performed. Trabecular surface finish was added to the reconstructed regions where osseointegration was desired. Finally, once a perfect fit was achieved, the ultimate 3D printed model was produced from the selected material—Ti-6Al-4V alloy and polyethylene. At this stage, in case of metal 3D printing, ultrasonic washing and oxidation followed and if necessary, impregnation surface coating with silver or hydroxyapatite was applied. Once the ultimate custom-made implant was complete, the non-metal-containing material was sterilized with ethylene oxide, avoiding exposure to high temperature. A final double layer sterile packaging was required for transfer to the operating room.

All four patients were followed up on a weekly basis during the first postoperative month and then two months after surgery via wound site control, X-rays, sedimentation, as well as inspection of C-reactive protein, leucocytes, red blood cells, hemoglobin, platelets and other routine biochemistry parameters. 

The following five parameters were included and weighed with up to 20 points each in the patient satisfaction survey: (1) functional result, (2) cosmetic result, (3) pain level, (4) self-assessment of the treatment process, and (5) pre-surgery waiting time. A total of 80–100 points were scored as an excellent patient satisfaction, 60–80 points—as a good one, 40–60 points—as a satisfactory one, and less than 40 points—as a poor satisfaction.

## 3. Results

Based on CT scanning, personalized 3D printed prostheses were designed, produced in a virtual environment and used in four patients within three to five days post injury or admission. The same workflow—as visualized in Figure 1—was used for all four clinical cases.

### 3.1. Clinical Case 1

A 3D printed custom-made prosthesis made of Ti-6Al-4V alloy was manufactured for a 41-year-old male patient with a comminuted radial head fracture with no possibility for primary plating (Figure 2). The stem of the prosthesis was made with a trabecular structure for better osseointegration. The surgery was performed three days post-trauma, after CT scanning of both left and right radial heads. During surgery, the range of motion (ROM) of both arms was evaluated. A full ROM of both forearms was achieved. The intraoperative C-arm control confirmed a perfect compatibility of the custom-made prosthesis with the native radial head characteristics. The patient was discharged with a long arm splint in supine position two days postoperatively. The splint was removed after two weeks, and mobilization was initiated. Full ROM of the affected elbow was measured at the two-year follow-up with patient’s satisfaction evaluated as excellent in the corresponding questionnaire (Table 2).

### 3.2. Clinical Case 2

An 18-year-old female patient was admitted to our clinic after a high-energy traffic accident, with the medial part of the distal humerus completely comminuted with a critical size bone defect. No bone substrate for primary fixation was available. A custom-made 3D printed prosthesis was designed and produced from Ti-6Al-4V alloy to accommodate the distal humerus defect (Figure 3). Considering the distal humerus trochlea fixation techniques according to the AO principles, the prosthesis provided sufficient trochlea integrity, connecting both medial and lateral columns. After determination of the missing zone, an extension to the humeral shaft similar to a medial plate was added to design a fixation bar that would connect both columns with locking head screws. Micro-trabecular finish was manufactured on the surface of the prosthesis in contact with the medullary canal of the humerus for better osseointegration.

The operation was performed five days after the trauma. During the surgery, the ulnar border was preserved by a standard 5 cm medial incision for implantation of the prosthesis in the bone defect region. All screws were inserted with pre-calculated lengths. During the fluoroscopic control, the joint motion was checked, and full ROM achieved. The patient was discharged with a long arm splint two days postoperatively. Passive mobilization was initiated in the third week post operation and the patient was referred to physical therapy. After finishing the physical therapy course, the early functional results were evaluated as good in the questionnaire (Table 2). In the early postoperative period, the patient’s extension was limited to 10 degrees while performing full flexion, supination and pronation. The daily activities could be fulfilled with no pain. Three years after surgery the patient experienced mild pain in the medial elbow, believed to be due to chondropathy and degeneration, which may have occurred as a result of poor compatibility between the implant material and cartilage. Currently, the patient is able to work with a little pain of the operated elbow. A total elbow joint prosthesis is planned if the pain increases and/or joint degeneration is detected.

### 3.3. Clinical Case 3

A personalized prosthesis was planned for a 58-year-old female patient with suspected tumor formation in the mandibular region, diagnosed later after biopsy as ameloblastoma. After examining the patient’s data, the osteotomy site was determined 3 cm distally to the tumor. A three-part prosthesis was designed: the first part made of Ti-6Al-4V alloy for the mandible, the second one made of Ti-6Al-4V alloy for the trochlear fossa, and the third one made of highly cross-linked polyethylene to act as a mandibular head at the joint between the metal parts (Figure 4). The prosthesis was attached to the mandible with a plate and fixed with a separate prosthetic plate to the trochlear fossa. For better osseointegration, the temporal part of the prosthesis contacting the surface of the mandible was produced with a micro-trabecular surface finish. The operation was performed four days after the planning phase. The surgical procedure included two incisions, one over the temporomandibular joint and one distally over the mandible. The temporomandibular joint was removed while preserving the facial nerve, and the mandible was resected from the distally planned osteotomy line. The three-part prosthesis was placed in the operative field and fixed with screws. The patient was on a liquid diet for three weeks after operation. Early-term patient satisfaction was evaluated as excellent. Currently, twelve months post operation, the patient can eat well, including solid meals.

### 3.4. Clinical Case 4

A 65-year-old male patient, who previously underwent four operations including total hip replacement and revision surgeries, was admitted with an acetabular defect. The affected acetabular site was too large for implantation of standard acetabular cages. Large defects may occur in the acetabulum after total hip arthroplasty operations, especially in patients who have undergone several revision surgeries. Although the gold standard for treatment of such defects considers use of acetabular cages, defects with severe bone loss cannot be easily addressed. In such cases, material is placed in the iliac wing via a method known as “ice cream cone prosthesis”, or the patient may have to live without a hip joint after removing all the material and undergoing an operation with the “Girdlestone technique”. After “ice cream cone” reconstruction, the biomechanics of the hip changes with decreased function and patient’s satisfaction. On the other hand, following Girdlestone surgery, the patient cannot bear loads on the operated side, which constitutes a major obstacle during daily living. Personalized 3D printed implants can be used very successfully in patients with such bone defects when standard devices are incapable to address the problem. The desired inclination and anteversion of the hip prosthesis, which is compatible with the liner and completely encompasses the defect, is ensured. The hip rotation center is calculated, and the location with the desired orientations of the planned screws are determined. The lever arm of the gluteus medius muscle can be adjusted to provide the necessary power for pelvis stabilization during walking, preventing Trendelenburg gait with close to normal physiology. Çıtak et al. reported good results with personalized 3D printed prostheses applied to large acetabular bone loss [7]. However, they did not report on either the application of a surface finish to increase the osseointegration of the prosthesis or the use of a silver coating to prevent infection.

A custom-made 3D printed prosthesis was designed and manufactured from Ti-6Al-4V alloy in accordance with a standard type of hip prosthesis (Figure 5). The surfaces with a contact to bone were designed with a trabecular structure to enhance osseointegration, while surfaces without such a contact were coated with silver ions to prevent infection. The patient was operated on the fifth day after admission using the previous incision site with a posterior Kocher approach to access the affected hip. Intraoperative insertion of the prosthesis into the acetabular defect was easy and fast. The joint’s center of rotation and leg length discrepancies were assessed during implantation and the surgical procedure was completed. Weight-bearing was initiated right after surgery and the physical therapy started two weeks later due to decreased muscle strength. Factor Xa inhibitor was administered to prevent deep vein thrombosis two months after surgery. The patient’s early-term satisfaction was evaluated as excellent in the questionnaire (Table 2). Two years postoperatively the patient was fully weight-bearing without the use of supportive devices and without feeling pain.

The functional results and patient satisfaction of the four clinical cases are summarized in Table 2.

## 4. Discussion

Although numerous standardized implant designs exist in the field of orthopedics and traumatology, they may be insufficient for treatment of some specific clinical cases [4,5]. In such cases, 3D printed implants can be used to address a variety of pathologies that would otherwise be challenging to manage with medical devices made via traditional manufacturing [44]. Moreover, the latter tends to require a central manufacturing site with space to store large inventories. In contrast, on-demand manufacturing, made possible via 3D printing, has changed this workflow and eliminated the need for a large production and storage space. Therefore, 3D printing of patient specific implants, instruments and anatomic models has impacted patient care and education in a variety of orthopedic subspecialties [5,45,46,47,48,49].

We believe that the personalized 3D printed radial head prosthesis in our first case is more effective than standard radial head prostheses as the radial head is not completely circular and since it is the center of forearm movement and rotation, it is difficult to achieve full anatomical compatibility when using standard radial head prostheses [50,51]. Anatomically precise placement at such an important point of motion is significant and the correct implant size is an important factor to avoid subluxation of the radial head [52]. The current published studies have reported that the highest incidence of removal/revision of radial head prostheses occurs within 2 years after implantation [53].

Our second case, presented with a large bone defect in the medial distal humerus, is rather rarely encountered. Osteosynthesis is the primary choice for treatment of distal humerus medial fractures. Unfortunately, this 18-year-old patient did not have enough bone fragments for fixation, though arthrodesis or total elbow prosthesis was possible. However, since the functional results in total elbow prostheses are not very satisfactory [54] and the lateral portion of the patient’s elbow was completely intact, we decided to design a personalized prosthesis. The patient did not have ulnar nerve damage and achieved almost complete ROM with extension limited to 10 degrees and good outcome in the satisfaction questionnaire. Two years postoperatively the joint was described as painless and stable. However, since the medial part of the distal humerus is a hinge-shaped joint, the metal fragment may have interacted slightly with the olecranon during load-sharing. Although no degeneration was observed on the X-rays in the third year, the fact that the first two years were painless, and that mild pain started in the third year led us to believe that there might be incompatibility between the metal and cartilage that may cause future joint degeneration. Then, a total joint replacement may be planned if progressive joint degeneration occurs.

In our third case we worked in collaboration with maxillofacial surgeons. In such procedures, a bone resorption can be observed in allografts, especially after radiotherapy in the postoperative period [55]. The plates used are not sufficiently strong and ultimate solutions for the temporomandibular joint cannot be found. Temporomandibular joint prostheses are often too short and cannot be used in such cases where custom-designed prostheses have demonstrated good results [56]. In this patient, chewing was gradually initiated after 20 days of postoperative stabilization and liquid diet. Swallowing and chewing exercises were prescribed. The patient, now in the first year after surgery, can open and close the mouth and can eat solid foods without experiencing jaw asymmetry. The speech is unaffected and functions close to the normal standards were achieved. No facial nerve damage or other complications were observed. The early patient satisfaction questionnaire was evaluated as excellent.

In our fourth case, the patient who had previously undergone four hip joint surgeries with no signs of infection, was admitted to our clinic with a limb length discrepancy of the affected leg, inability to bear weight and weakness due to muscle atrophy. Several options were evaluated during preoperative planning. We could have planned an operation using an “ice-cream cone prosthesis” with a stem placed in the iliac wing [57]; however, the anatomically high center of rotation and limb length discrepancy—with risks of instability and recurrent dislocation—led us to the decision of applying a patient-specific 3D printed prosthesis. In such cases, adequate stabilization is achieved by complete closing of the defect and placing the desired number of screws in the proper locations [2]. In addition, since the joint rotation center is kept in its anatomical position, the function is better, with no leg-length discrepancy. Full weight-bearing was initiated immediately post operation, both legs were equal in length and an excellent score was registered in the early-term satisfaction questionnaire. Unrestricted full weight-bearing normal daily activities continued in the second postoperative year. The clinical results of this case are in agreement with the improved stability, decreased pain and better outcomes reported in a previous clinical study on patients undergoing revision of an acetabular defect with use of 3D printed titanium acetabular cups versus standard implants used in the control group, where the authors hypothesized that the increased porosity and homogenous aperture of the 3D printed implants facilitated bone growth better than traditionally manufactured acetabular cups [58].

Generally, custom-made prostheses are the ultimate solution in modern reconstructive orthopedic surgery, with very good results, especially in patients with critical size bone defects [7]. Due to the low infection rate resulting from the silver coating [29] and the designed trabecular implant contact surfaces, a shortened osseointegration time can be expected [27]. Personalized 3D printed prostheses achieve highly functional results due to (1) precise anatomical bone replacement, (2) the possibility to generate complex free-form interconnected implant surfaces with a potential to facilitate osseointegration, reduce stiffness mismatch at bone–implant interfaces and address the stress-shielding problem, and (3) increased conformity between implants and patients’ anatomical needs [4,5,59,60].

Although the potential clinical performance of 3D printed implants can be outstanding due to their ability to address reconstructive challenges beyond the scope of off-the-shelf implants, some limitations of this emerging technology should be noted, including high implant costs, extended time needed for scheduling, designing and manufacturing, some regulatory considerations that remain challenging for hospitals as currently there are limited standards to monitor the safety of 3D printed custom products, and lack of intraoperative flexibility [5,61]. In addition, CT scanning of the contralateral side during preoperative planning exposes the patient to an excess of radiation. Therefore, custom-made implants should be used only in carefully selected cases.

Overall, although the clinical data supporting the routine use of 3D printed implants are currently limited, the integration of the 3D printing technology with computer-assisted navigation and implant placement is very promising as it may further enhance the performance of personalized implants in patient-specific orthopedics. Further studies and more frequent applications over time will allow for design optimizations and evidence-based results with diminished bias to investigate the real efficacy of the 3D printed implant use in the clinical practice.

## 5. Conclusions

Personalized 3D printed prostheses offer an opportunity for a treatment of choice and provide good anatomical and functional results, shortened surgical time, less complications, and high satisfaction in patients with appropriate indications. The method should be considered primarily for patients with large bone defects, or such indicated for resection. Personalized 3D printed prostheses have the potential to become more common and beneficial in the future.

## Figures and Tables

**Figure 1 medicina-59-00249-f001:**
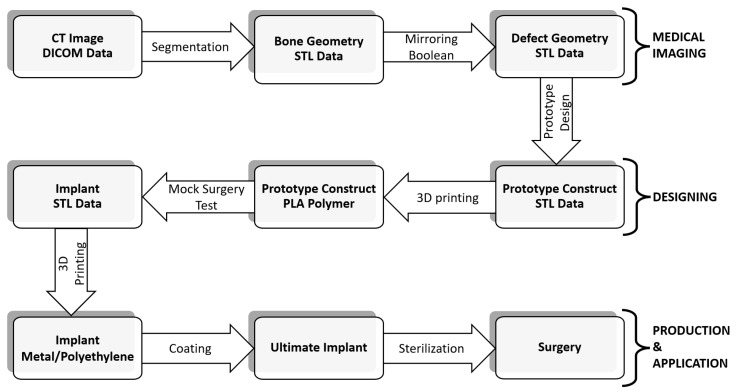
Workflow for medical imaging, designing, prototyping, production and surgical application of personalized 3D printed implants.

**Figure 2 medicina-59-00249-f002:**
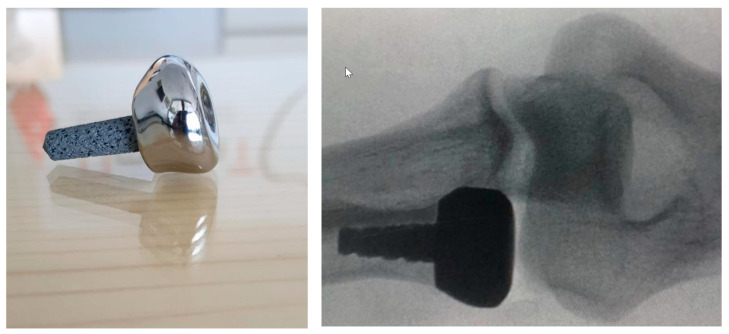
A 3D printed custom-made radial head prosthesis (**left**) and an X-ray of its implantation (**right**).

**Figure 3 medicina-59-00249-f003:**
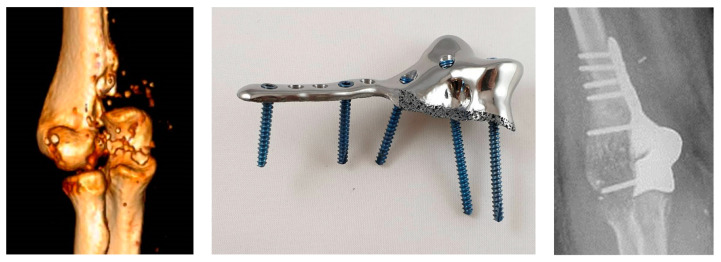
A CT reconstruction visualizing the completely comminuted medial part of a distal humerus with a critical size bone defect (**left**) together with a 3D printed custom-made prosthesis designed for the distal humerus defect (**middle**) and an X-ray of its implantation (**right**).

**Figure 4 medicina-59-00249-f004:**
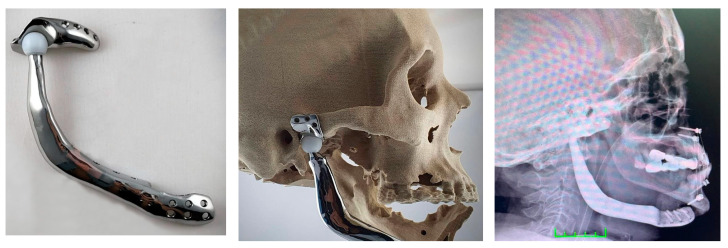
A 3D printed three-part custom-made prosthesis designed to address suspected tumor formation in the mandibular region of a patient (**left**) together with its trial mounting on a 3D printed model (**middle**) and an X-ray of its implantation (**right**).

**Figure 5 medicina-59-00249-f005:**
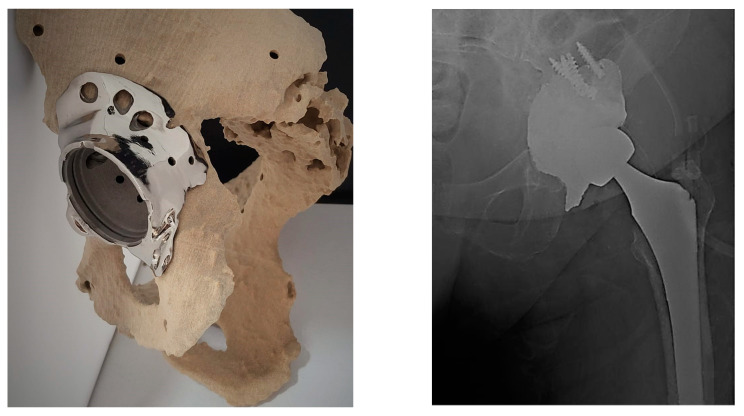
A 3D printed custom-made prosthesis designed to address an acetabular defect of a patient and mounted on the 3D printed model of the patient’s pelvis (**left**), together with an X-ray of its implantation (**right**).

**Table 1 medicina-59-00249-t001:** Characteristics of the patients included in the study.

Case	Age (Years)	Sex	Weight (kg)	Cardiovascular Disease	Diabetes	Smoking	Other Disease
1	41	M	78	No	No	Yes	Not known
2	18	F	51	No	No	No	Not known
3	58	F	85	No	No	No	Not known
4	65	M	81	Hypertension	No	No	Asthma

**Table 2 medicina-59-00249-t002:** Functional results and patient satisfaction of the four clinical cases.

Case	Localization	Functional Results	Patient Satisfaction
1	Radial head	Full ROM in elbow & forearm	Excellent
2	Distal humerus	140° flexion 10° restricted extension Full supnation and pronation	Good
3	Mandibula	Can speak and eat easily Symmetrical mouth opening and closing	Excellent
4	Acetabulum	No leg length discrepancy 100° flexion 10° extension 20° internal rotation 30°external rotation 40° abduction 10° adduction Painless walking with full weight-bearing	Excellent

## Data Availability

The datasets analyzed during the current study are available from the corresponding author upon reasonable request. In order to comply with the requirements of the Ethics Committee, the image set is not available for request due to data privacy policies.

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
