# Peer review of "The Possibilities of Personalized 3D Printed Implants—A Case Series Study"

_medicina, 2023, doi:10.3390/medicina59020249_

Round 1
Reviewer 1 Report
General comments
The article The Possibilities of Personalized 3D Printed Implants – A Case Series Study is well written and well organized. The research topic is original and addresses an important issue. The purpose of this study is to present the design, modeling and manufacturing stages of four different personalized 3D printed prostheses and their application to clinical cases of patients who have undergone surgery in different anatomical locations with precisely defined indications for implantation. The methodology of the adopted method was described in a concise and clear manner. The results have been discussed in a correct manner allowing for easy interpretation. The discussion presented in the paper was written correctly, however, the authors should refer to more papers on similar topics. Conclusions are presented in a concise manner and are related to the results obtained. The list of references contains only 26 items which is far too small a number for a scientific paper. In my opinion, it seems necessary to make minor corrections to allow a more thorough understanding of the topic. The following are my comments.
Minor comments:
Introduction
The introduction should be expanded to include information on typical applications of Computer-Assisted Orthopedic Surgery along with the necessary literature. The authors should also add information on the potential applications of 3D models developed from DICOM files and the finite element method in musculoskeletal biomechanics and orthopedics. Below are some literature items where authors will find useful information:
doi:10.3389/fsurg.2015.00066; doi:10.1016/j.cuor.2007.03.002; DOI10.1051/matecconf/201925207007; DOI10.1051/itmconf/20171507015; DOI10.1051/itmconf/20171507006; DOI10.12913/22998624/64064; DOI10.3390/ma15144731
Materials and Methods
The authors should supplement this chapter with more detailed information on the process of creating 3D models from documentation in DICOM format. More detailed information on the FEA analyses performed should also be added. Please consider rebuilding this chapter and possibly dividing it into subsections.
Results
The results in for all four cases were described correctly. It is worth expanding with additional graphics from the design process and FEA analyses.
Discussion
The authors should add to the discussion a brief introduction on the potential of Computer-Assisted Orthopedic Surgery along with the corresponding literature (I have provided suggestions above). I ask that the discussion be expanded to include the results described in papers on similar topics presented by other authors. The authors add more detailed information on the limitations of the proposed method, the risks of using 3D printed implants and future plans.
After making the appropriate corrections, the article can be accepted for publication.
Author Response
The article The Possibilities of Personalized 3D Printed Implants – A Case Series Study is well written and well organized. The research topic is original and addresses an important issue. The purpose of this study is to present the design, modeling and manufacturing stages of four different personalized 3D printed prostheses and their application to clinical cases of patients who have undergone surgery in different anatomical locations with precisely defined indications for implantation. The methodology of the adopted method was described in a concise and clear manner. The results have been discussed in a correct manner allowing for easy interpretation. The discussion presented in the paper was written correctly, however, the authors should refer to more papers on similar topics. Conclusions are presented in a concise manner and are related to the results obtained. The list of references contains only 26 items which is far too small a number for a scientific paper. In my opinion, it seems necessary to make minor corrections to allow a more thorough understanding of the topic. The following are my comments.
Reply: Thank you very much for this summary.
Minor comments:
Introduction
The introduction should be expanded to include information on typical applications of Computer-Assisted Orthopedic Surgery along with the necessary literature. The authors should also add information on the potential applications of 3D models developed from DICOM files and the finite element method in musculoskeletal biomechanics and orthopedics. Below are some literature items where authors will find useful information:
doi:10.3389/fsurg.2015.00066; doi:10.1016/j.cuor.2007.03.002; DOI10.1051/matecconf/201925207007; DOI10.1051/itmconf/20171507015; DOI10.1051/itmconf/20171507006; DOI10.12913/22998624/64064; DOI10.3390/ma15144731
Reply: Thank you very much for this recommendation. Five corresponding paragraphs were added in Introduction implementing new literature as recommended.
Materials and Methods
The authors should supplement this chapter with more detailed information on the process of creating 3D models from documentation in DICOM format. More detailed information on the FEA analyses performed should also be added. Please consider rebuilding this chapter and possibly dividing it into subsections.
Reply: Thank you for this remarks. The Materials and Methods part was correspondingly amended and Figure 1 was replaced with an optimized one. As the designing of the patient-specific implants was based on adopted geometry of standardized and already biomechanically tested existing implants and prostheses (used in the starting step without any weakening), only implant fitting to the recipient bone site (affected bone) with no FEA was tested and optimized.
Results
The results in for all four cases were described correctly. It is worth expanding with additional graphics from the design process and FEA analyses.
Reply: Thank you for this remark. As mentioned in the reply above, the designing of the patient-specific implants was based on adopted geometry of standardized and already biomechanically tested existing implants and prostheses used in the starting step without any weakening. Therefore, only implant fitting to the recipient bone site (affected bone) with no FEA was tested and optimized.
Discussion
The authors should add to the discussion a brief introduction on the potential of Computer-Assisted Orthopedic Surgery along with the corresponding literature (I have provided suggestions above). I ask that the discussion be expanded to include the results described in papers on similar topics presented by other authors. The authors add more detailed information on the limitations of the proposed method, the risks of using 3D printed implants and future plans.
Reply: Thank you for this remark. Five paragraphs were added in Discussion in this regard.
After making the appropriate corrections, the article can be accepted for publication.
Reply: Thank you very much.
Reviewer 2 Report
In the manuscript are presented 4 clinical cases in which patients were subjected to implantation of 3D bioprinted prosthesis. The manuscript presents the data in a clear, synthetic manner. Please add the following:
- in the materials and method section add a table with patients data: age, sex, weight, cardiovascular disease, diabetes, smoking.....
- the material used for 3D printing should be mentioned in the material and method section
- after implantation of the prosthesis were performed blood/hematological test to see the number of leucocyte, thrombocyte, erytrocyte, LHD, interleukins, etc ?
- how long after implantation were the observation/test performed? Please add in the manuscript.
Author Response
In the manuscript are presented 4 clinical cases in which patients were subjected to implantation of 3D bioprinted prosthesis. The manuscript presents the data in a clear, synthetic manner.
Reply: Thank you for this summary.
Please add the following:
- in the materials and method section add a table with patients data: age, sex, weight, cardiovascular disease, diabetes, smoking.....
Reply: Thank you for this recommendation. These patients' data were added in Table 1.
- the material used for 3D printing should be mentioned in the material and method section
Reply: Thank you. The materials for 3D printing of the prototype (PLA) and ultimate implants (Ti-6Al-4V alloy and polyethylene) were added in Methods.
- after implantation of the prosthesis were performed blood/hematological test to see the number of leucocyte, thrombocyte, erytrocyte, LHD, interleukins, etc ?
Reply: Yes, blood/hematological tests were performed. This information is added in the Methods now.
- how long after implantation were the observation/test performed? Please add in the manuscript.
Reply: Thank you. This information was added in the manuscript text.
Reviewer 3 Report
General Comment:
This work present an important example of the application of personalized 3D printed implants in orthopedic surgery. The authors developed an ideal workflow of the procedure which will benefit from the addition of detailed information regarding each step to have a strong impact in the field. Moreover, this study could have been designed to include a comparison with results of pre-made implants applied to similar anatomical sites, to better underline the advantages of the 3D personalized implant.
Introduction:
Page 2 Line 56: “high patient satisfaction” I suggest including some reference regarding the benefit of personalized implants respect to standard ones and how they impact specifically on, for example, pain, recovery time, integration, rejection, and function.
Page 2 Line 58: “The design of personalized implants depends entirely on the surgeon's knowledge and the patient's needs” I suggest extending this sentence considering the multidisciplinary nature of the personalized implant design as depicted in Figure 1 and to better clarify the specific role of the surgeon in the design process.
Page 2, Line 67: “thes” should read “these”
Materials and Methods:
The material and method section could be improved by adding extended details regarding the methodologies used such as: the hardware and parameters used to perform the Computed Tomography; the software used to generate the STL files from the DICOM images; the software used to perform FEA; the redesign approach, the polymer material used for the matching test, and the identity of the final material of the implant for each application. Moreover, the authors should clarify if the same approach was used for the 4 different applications.
Figure 1: “Matterial” should read Material
Results:
In general, a comparative analysis between the outcomes (such as the ones presented in table 1) of 3D personalized implants produced with the workflow described by the authors and the results of pre-made implants, or alternatively of other personalized implants designed with a different workflow applied to similar anatomical sites, could strengthen the impact of the present work.
Table 1: The results will benefit from the inclusion of the parameters used in the questionnaire to determine the patient’s satisfaction rate
Author Response
This work present an important example of the application of personalized 3D printed implants in orthopedic surgery. The authors developed an ideal workflow of the procedure which will benefit from the addition of detailed information regarding each step to have a strong impact in the field. Moreover, this study could have been designed to include a comparison with results of pre-made implants applied to similar anatomical sites, to better underline the advantages of the 3D personalized implant.
Reply: Thank you very much for this summary.
Introduction:
Page 2 Line 56: “high patient satisfaction” I suggest including some reference regarding the benefit of personalized implants respect to standard ones and how they impact specifically on, for example, pain, recovery time, integration, rejection, and function.
Reply: Thank you for this recommendation. Both Introduction and Discussion were extended in this regard.
Page 2 Line 58: “The design of personalized implants depends entirely on the surgeon's knowledge and the patient's needs” I suggest extending this sentence considering the multidisciplinary nature of the personalized implant design as depicted in Figure 1 and to better clarify the specific role of the surgeon in the design process.
Reply: Thank you for this recommendation. A sentence was added in this regard.
Page 2, Line 67: “thes” should read “these”
Reply: Thank you. This is corrected now.
Materials and Methods:
The material and method section could be improved by adding extended details regarding the methodologies used such as: the hardware and parameters used to perform the Computed Tomography; the software used to generate the STL files from the DICOM images; the software used to perform FEA; the redesign approach, the polymer material used for the matching test, and the identity of the final material of the implant for each application. Moreover, the authors should clarify if the same approach was used for the 4 different applications.
Reply: Thank you for this recommendation. This information was added in Methods. As the the designing of the patient-specific implants was based on adopted geometry of standardized and already biomechanically tested existing implants and prostheses used in the starting step without any weakening, only implant fitting to the recipient bone site (affected bone) with no FEA was tested and optimized.
Figure 1: “Matterial” should read Material
Reply: Thank you. Figure 1 was replaced with an optimized one considering this correction.
Results:
In general, a comparative analysis between the outcomes (such as the ones presented in table 1) of 3D personalized implants produced with the workflow described by the authors and the results of pre-made implants, or alternatively of other personalized implants designed with a different workflow applied to similar anatomical sites, could strengthen the impact of the present work.
Reply: Thank you for this recommendation. The results for some of the cases were compared with existing literature upon the availability of the latter.
Table 1: The results will benefit from the inclusion of the parameters used in the questionnaire to determine the patient’s satisfaction rate.
Reply: Thank you for this recommendation. The parameters are added in Methods.
Round 2
Reviewer 3 Report
Tha authours addressed the main comments improving the methodology section and effecting the deliverability of the developed workflow.